# Mechanic-Electric-Thermal Directly Coupling Simulation Method of Lamb Wave under Temperature Effect

**DOI:** 10.3390/s22176647

**Published:** 2022-09-02

**Authors:** Xiaofei Yang, Zhaopeng Xue, Hui Zheng, Lei Qiu, Ke Xiong

**Affiliations:** Research Center of Structural Health Monitoring and Prognosis, State Key Laboratory of Mechanics and Control of Mechanical Structures, Nanjing University of Aeronautics and Astronautics, Nanjing 210016, China

**Keywords:** structural health monitoring, Lamb Wave, multiphysics simulation, temperature effect, thermal stress

## Abstract

Lamb Wave (LW)-based structural health monitoring method is promising, but its main obstacle is damage assessment in varying environments. LW simulation based on piezoelectric transducers (referred to as PZTs) is an efficient and low-cost method. This paper proposes a multiphysics simulation method of LW propagation with the PZTs under temperature effect. The effect of temperature on LW propagation is considered from two aspects. On the one hand, temperature affects the material parameters of the structure, the adhesive layers and the PZTs. On the other hand, it is considered that the thermal stress caused by the inconsistency of thermal expansion coefficients among the structure, the adhesive layers, and the PZTs affect the piezoelectric constant of the PZTs. Based on the COMSOL Multiphysics, the mechanic–electric–thermal directly coupling simulation model under temperature effect is established. The simulation model consists of two steps. In the first step, the thermal-mechanic coupling is carried out to calculate the thermal stress, and the thermal stress effect is introduced into the piezoelectric constant model. In the second step, mechanic–electric coupling is carried out to simulate LW propagation, which considers the piezoelectric effect of the PZTs for the LW excitation and reception. The simulation results at −20 °C to 60 °C are obtained and compared to the experiment. The results show that the *A*_0_ and *S*_0_ mode of simulation signals match well with the experimental measurements. Additionally, the effect of temperature on LW propagation is consistent between simulation and experiment; that is, the amplitude increases, and the phase velocity decreases with the increment of temperature.

## 1. Introduction

Structural Health Monitoring (SHM) technology has the advantages of real-time monitoring, reducing detection times and improving detection efficiency and condition-based maintenance. It has been widely studied and applied in the field of aerospace [1,2,3]. The Lamb Wave (LW)-based SHM method is a promising one, which has the advantages of high sensitivity and large area monitoring through piezoelectric transducers (PZTs) network. The method’s fundamental principle is to directly attach PZTs to a monitored structure and acquire LW signals via the excitation and sensing of these PZTs. The damage can be measured by analyzing the LW features that the damage has changed. Therefore, it shows great application potential in structural damage assessment [4,5,6]. However, in recent decades, the development of the LW-based SHM method from theoretical and basic research to engineering application has been quite slow [7,8,9].

Time-varying conditions are one of the main obstacles to the development of LW-based SHM [10]. When LW propagates in aircraft structures, it will inevitably be affected by time-varying conditions, including temperature, moisture, load, and aerodynamic noise. The extracted damage features are affected by these environmental factors, which reduce the reliability of damage monitoring. A variety of damage monitoring methods have been developed, such as baseline free method, environmental compensation method and probabilistic model method, to minimize the impact of time-varying conditions [11,12,13,14,15]. In order to further study and verify these damage monitoring methods in aerospace applications, a large number of LW signals must be obtained under time-varying conditions.

Although the LW signals under time-varying conditions can be obtained by experiment and simulation, respectively, it is expensive and time-consuming to obtain LW signals via experiment—especially, those experiments carried out on real aircraft structures under in-service conditions. However, the LW simulation under time-varying conditions is an effective and low-cost method, which can not only study the LW propagation in an aircraft structure but also can be used to verify the related damage monitoring methods.

In recent decades, Finite Element Analysis (FEA)-based LW simulation methods [16,17,18,19,20,21,22] have been intensively explored for simulating LW propagation in metallic and composite structures. However, the LW simulation method under time-varying conditions considering the PZTs attached to structures is rarely reported.

The varying temperature condition is a significant one. It has been studied that the temperature can cause significant changes in the velocity and amplitude of LW signals acquired by the PZTs [23,24]. The influence of temperature on LW propagation is studied via theoretical studies and numerical analysis [25,26,27,28]. The results show that the main reason for the variations of the LW velocity and amplitude under the temperature effect is due to the change in structural materials and the piezoelectric properties, including the piezoelectric constant and dielectric constant. The effect of temperature change on the LW is investigated over a temperature range of −200 °C to 204 °C via ABAQUS [29]. Since the influence of temperature on material parameters is not considered, temperature change in the range of −200 °C to 93 °C has no effect on the displacement responses. Attarian et al. [30] experimentally investigated that thermal cycling reduces the sensitivity of damage diagnosis because the properties of adhesive layers have changed and how the influence of the adhesive layers is not negligible. So far, the influence of thermal stress caused by the inconsistency of thermal expansion coefficients among different materials on LW propagation has not been studied.

To simulate the temperature influence on LW, Lonkar et al. [31] studied the piezo-enabled spectral element analysis method containing a piezoelectric model for LW propagation. In the simulation model, the piezoelectric constant and dielectric constant of the piezoelectric sheet and the shear modulus, Poisson’s ratio and elastic modulus of the adhesive layers are considered as a function of temperature. Palazotto et al. [32] developed a 2D simulation model containing a piezoelectric sheet and investigated the influence of temperature on LW on aluminum plates using ABAQUS simulation software. The influence of temperature on the elastic modulus and Poisson’s ratio of aluminum plate is considered in the simulation model. The results show that LW propagation velocity decreases with the increment of temperature. Yule et al. [33] conducted a 2D-guided wave simulation based on COMSOL Multiphysics software, considering the influence of temperature on material parameters, and the influence law of temperature on LW was obtained. However, these studies neglected the influence of thermal stress caused by the inconsistency of thermal expansion coefficients among different materials on LW propagation. The piezoelectric constant of PZT is very sensitive to thermal stress. Thermal stress leads to the change of the piezoelectric constant of PZT, which will affect LW propagation. Therefore, thermal stress needs to be considered.

This paper proposes a multiphysics simulation method for LW propagation with the PZTs under temperature effect. The simulation model includes the changes in material parameters of the structure, the adhesive layers, and the PZTs with temperature. In particular, the thermal stress that affects the piezoelectric constant due to the inconsistency of the thermal expansion coefficients among the three is considered. The FEA model of LW propagation with the PZTs under temperature effect is constructed based on the COMSOL Multiphysics computational platform. The LW signals under the temperature effect can be derived directly from the simulation model. The results show that the waveform of simulation signals matches well with the experimental measurements, which indicates that the simulation method is feasible.

## 2. Simulation Mechanisms of Temperature Influence on LW

Take the plate-like structure as an example to illustrate the simulation mechanisms of LW under the temperature effect [34], as shown in Figure 1. Two PZTs are attached to the structure with an adhesive and are used as an LW actuator and LW sensor, respectively. A voltage excitation waveform is provided to the actuator to generate the LW signal. The actuator deforms and transmits the stress to the adhesive layers and the structure due to the inverse piezoelectric effect. When LW propagates to the sensor in the structure, it is received and converted into a voltage signal due to the direct piezoelectric effect. It is proposed to simplify the problem of temperature affecting LW propagation into two parts. One part is that temperature affects the material parameters of the structure, adhesive layers, and PZTs. The other part is that thermal stress affects the piezoelectric constant of the PZTs due to the inconsistency of thermal expansion coefficients among the structure and the adhesive layers, and the PZTs. The two parts are superimposed for LW propagation simulation under the temperature effect. In this Section, two parts of the influence mechanisms are introduced, and the numerical model used in this paper is established.

### 2.1. Excitation–Propagation–Sensing Model

#### 2.1.1. Excitation Model

When the PZT is employed as an excitation element, the PZT’s principle for LW excitation is the inverse piezoelectric effect. The piezoelectric constitutive equations are presented in the following Equation (1).
(1)$$\begin{array}{l} e=d^{T}Ε+s\sigma \\ D=\varepsilon Ε+d\sigma \end{array}$$
where ***e*** and ***E*** refer to strain vector and electric field vector, respectively, ***s*** is the elastic compliance matrix, ***σ*** is the stress vector, ***ε*** is the dielectric constant matrix, and ***d*** and ***D*** refer to the piezoelectric coefficient matrix and electric displacement vector, respectively.

According to Giurgiutiu’s study [35], the strain transfer model between the actuator and the structure is derived by converting the input voltage to the mechanical strain and then by calculating the shear-lag model between the top and lower adhesive layer interfaces. The schematic diagram of LW excitation is shown in Figure 2.

Where *l*_act_ and *t*_act_ are the diameter and thickness of the PZT, respectively, *t*_bond_ is the thickness of the adhesive layer, *t*_plate_ is the thickness of the plate structure, *G*_bond_ is the shear strength of the adhesive layer, $Y_{\mathrm{act}}^{E}$ is Young’s modulus of the PZT, and *E*_plate_ is elastic modulus of the structure.

In this paper, we use a circular PZT with *d*_31_ = *d*_32_ and excite the LW by applying a voltage excitation signal in the 3-direction. Only the piezoelectric constant *d*_31_ needs to be considered in m/V. The driving strain at the bottom of the excitation sensor along the x-direction is expressed in the following Equation (2), where *V*_in_ is the excitation voltage.
(2)$$\varepsilon_{\mathrm{act}}(t)=-\frac{d_{31}V_{\mathrm{in}}(t)}{t_{\mathrm{act}}}$$

Based on the shear-lag model, the stress generated by the PZT coupled to the structure through the adhesive layer is shown in Equation (3).
(3)$$\tau \left(x\right)=-\frac{G_{\mathrm{bond}}}{t_{\mathrm{bond}}}\times \frac{l_{\mathrm{act}}\varepsilon_{\mathrm{act}}}{\Gamma \cosh\Gamma }\sinh\left[\Gamma \left(\frac{2x}{l_{\mathrm{act}}}\right)\right]$$
where Γ is the shear-lag coefficient, and the expression is shown in Equation (4). When *α* = 1, shear excitation is used to excite *S* mode. When *α* = 3, bending excitation is used to excite *A* mode.
(4)$$\Gamma^{2}=\frac{G_{\mathrm{bond}}l_{\mathrm{act}}^{2}}{t_{\mathrm{bond}}}\left(\frac{1}{Y_{\mathrm{act}}^{E}t_{\mathrm{act}}}+\frac{\alpha }{E_{\mathrm{plate}}t_{\mathrm{plate}}}\right)$$

#### 2.1.2. Propagation Model

For the plate structure with a free surface, the displacement and stress in the isotropic plate can be simplified into the LW frequency equation in symmetric and antisymmetric mode without consideration for in-plate stress, as shown in Equations (5) and (6).
(5)$$\frac{\tan(qh)}{\tan(ph)}=-\frac{4k^{2}pq}{{\left(q^{2}-k^{2}\right)}^{2}}$$
(6)$$\frac{\tan(qh)}{\tan(ph)}=-\frac{{\left(q^{2}-k^{2}\right)}^{2}}{4k^{2}pq}$$
where *p* and *q* are given by
(7)$$p^{2}=(\frac{\omega^{2}}{c_{L}^{2}}-k^{2}),\;q^{2}=(\frac{\omega^{2}}{c_{T}^{2}}-k^{2})\;$$
where *h* is half the thickness of the plate, *k* is wave number, *ω* is angel frequency, *c*_L_ and *c*_T_ are the velocities of the longitudinal wave and transverse wave, respectively, as shown in Equations (8) and (9).
(8)$$c_{L}=\sqrt{\frac{Y_{\mathrm{plate}}^{E}(1-\nu_{\mathrm{plate}})}{\rho_{\mathrm{plate}}(1+\nu_{\mathrm{plate}})(1-2\nu_{\mathrm{plate}})}}$$
(9)$$c_{T}=\sqrt{\frac{Y_{\mathrm{plate}}^{E}}{2\rho_{\mathrm{plate}}(1+\nu_{\mathrm{plate}})}}$$

#### 2.1.3. Sensing Model

Assuming that there is no loss in LW propagation, the strain at the sensor is obtained according to the basic equation in the adhesive shear-lag model, and the voltage output can be obtained according to the direct piezoelectric effect [26], as shown in Equation (10).
(10)$$V_{\mathrm{out}}\left(t\right)=d_{31}^{\mathrm{act}}C_{\mathrm{act}}\left(\Gamma \right)C_{\mathrm{sen}}\left(\Gamma \right){\left[\frac{d_{31}}{e_{33}s_{13}\left(1-\nu_{\mathrm{act}}\right)}\right]}_{\mathrm{sen}}V_{\mathrm{in}}\left(t\right)$$
where *ν*_act_ is the Poisson’s ratio of the PZT, *e*_33_ is the dielectric constant of the PZT, *s*_13_ is the elastic coefficient of the PZT, and *C*(Γ) is a function of the Γ shear-lag parameter, as shown in Equations (11) and (12).
(11)$$C_{\mathrm{act}}\left(\Gamma \right)=\frac{G_{\mathrm{bond}}\left(1+\nu_{\mathrm{act}}\right)R}{t_{\mathrm{bond}}\left[\left(\Gamma R\right)I_{0}\left(\Gamma R\right)-\left(1-\nu_{\mathrm{act}}\right)I_{1}\left(\Gamma R\right)\right]}I_{1}\left(\Gamma R\right)$$
(12)$$C_{\mathrm{sen}}\left(\Gamma \right)=\frac{\iint \left({\overset{⌢}{\varepsilon }}_{rr}^{\mathrm{sen}}+{\overset{⌢}{\varepsilon }}_{\theta \theta }^{\mathrm{sen}}\right)rdrd\theta \;|{}_{\tau_{0}=1}}{\pi R^{2}}$$
where *R* is the radius of the PZT. *I*(Γ*R*) is the Bessel function.

The LW theoretical model and the related temperature factors that affect the LW propagation velocity and response amplitude are summarized in Table 1. On the one hand, the temperature affects the material parameters of the structure, the adhesive layer and the PZT. On the other hand, the thermal expansion coefficients of the three are inconsistent, so the PZT is affected by thermal stress, which changes the mechanic-to-electric conversion characteristics. This is equivalent to the change produced by the external load on the PZT. Since the PZT is sensitive to stress, the thermal stress caused by thermal expansion is not negligible, so the change in temperature is accompanied by the load effect.

### 2.2. Material Parameters under Temperature Effect

In this paper, we take 2024 aluminum alloy as an example. The elastic modulus and Poisson’s ratio of 2024 aluminum material measured by laser ultrasound was published by Sandia National Laboratories [36]. Elastic modulus decreases approximately linearly with the increasing temperature, and Poisson’s ratio increases approximately linearly with the increasing temperature, as shown in Equations (13) and (14).
(13)$$E_{\mathrm{plate}}\left(\Delta T\right)=73.5-0.06\times \Delta T$$
(14)$$v\left(\Delta T\right)=0.344+5.13\times 10^{-5}\times \Delta T$$

According to the parameter manual of a typical two-component epoxy adhesive, its working temperature range is −55 °C to 250 °C, and its storage modulus at room temperature is 3.61 Gpa with a thermal expansion rate of 54 × 10^−6^/°C. Barakat et al. [37] gave the curve of storage modulus with temperature. The numerical model for the variation of adhesive shear modulus with temperature is shown in Equation (15).
(15)$$E_{\mathrm{bond}}\left(\Delta T\right)=3.61-0.01\times \Delta T$$

Compared to metallic structures, piezoelectric materials contain both force and electrical properties, so the parameters affected by temperature are more complex. By reviewing the relevant literature, NASA reported some measurement results about the variation of PZT parameters with temperature [38]. From the reported results, the impedance, mechanic-electric coupling coefficient, and dielectric loss of PZT-5A can be approximately constant in the range of −55 °C to 100 °C, which can be neglected in the simulation. As the temperature increases, the dielectric constant and piezoelectric constant show a linear increasing trend, which needs to be considered in the simulation.

The numerical model of the piezoelectric constant and dielectric constant of PZT-5A was studied in the temperature range of −20 °C to 60 °C [34], as shown in Equation (16) and Equation (17), in agreement with the findings reported by NASA [38]. This numerical model is used for simulation, but it does not take into account the thermal stress effect.
(16)$$d_{31}(\Delta T)=-167.7-0.194\times \Delta T$$
(17)$$e_{33}(\Delta T)=2155+4.12\times \Delta T$$

### 2.3. Piezoelectric Constants under Thermal Stress Effect

The thermal expansion coefficient of the PZT, adhesive layer, and aluminum structure are 3 × 10^−6^/°C, 54 × 10^−6^/°C, and 23 × 10^−6^/°C, respectively. The thermal stress is not negligible for the PZT because the difference between the three thermal expansion coefficients is large, and the piezoelectric sensor is sensitive to changes in stress.

For most materials, warming causes expansion and cooling causes contraction, and tiny cells generate thermal stress due to the constraints of adjacent cells and boundary conditions. It is common to assume a linear relationship between strain and temperature, as shown in Equation (18), which forms a set of intrinsic relationships between strain and temperature. The coefficient of thermal expansion is a fundamental parameter of the material, and the effect of thermal stress can be approximated as a PZT subjected to a static load.
(18)εx=εy=εz=αΔT

When a PZT is subjected to an external load, its polarization state changes. The internal electric dipole moments are aligned in the direction of the polarization field and are confined by the domain walls, thus changing the mechanic-electric transition characteristics of the PZT.

In the application of excitation-response of LW, the mechanic-electric transition characteristics of piezoelectric materials are usually considered to be linear. However, in fact, the piezoelectric material itself is nonlinear, and its mechanic-electric characteristics are usually reflected as a hysteresis curve, which can be approximated as linear due to the small voltage and displacement of the excitation and response of LW.

Qiu et al. [39] studied the variation law of LW amplitude and propagation velocity under static load. It was pointed out that LW propagation velocity is affected by load due to the acoustoelastic effect, and the structure has a nonlinear change in its stress-strain intrinsic relationship under external load. Since the effect of thermal stress on the structure is ignored when discussing thermal stress, the acoustoelastic effect is not considered in this paper. The amplitude is affected by the load because the piezoelectric constant of the PZT is changed. This paper focuses on the effect of thermal stress on the LW amplitude. The nonlinear numerical relationship model between the load and the piezoelectric constant *d*_31_ is summarized as shown in Equation (19), where *σ* is the actual stress caused by the load in MPa.
(19)$$d_{31}\left(\Delta \sigma \right)=d_{31}+d_{31}\times \left(-1.1\times 10^{-5}\times \Delta \sigma^{2}+4.2\times 10^{-3}\times \Delta \sigma \right)$$

In summary, the numerical model of the influence of static load on the piezoelectric constant studied by scholar Qiu is used to simulate the influence of thermal stress on the piezoelectric constant. The numerical models of the influence of temperature on the piezoelectric constant *d*_31_ and the influence of thermal stress on the piezoelectric constant *d*_31_ are superimposed as the numerical model of the temperature-influenced LW propagation simulation, including thermal stress, as shown in Equation (20).
(20)$$d_{31}(\Delta T,\;\Delta \sigma_{T})=-167.7-0.194\times \Delta T-167.7\times \left(-1.1\times 10^{-5}\times \Delta \sigma_{T}{}^{2}+4.2\times 10^{-3}\times \Delta \sigma_{T}\right)$$
where Δ*T* is relative to the reference temperature 20 °C, in °C, Δ*σ_T_* is the thermal stress, in MPa.

## 3. Simulation Method of LW under Temperature Effect

### 3.1. Architecture of the Multiphysics Simulation Method

Three kinds of physics need to be coupled with each other to simulate the LW under temperature effect. One is mechanic-electric coupling controlled by a piezoelectric instantaneous equation, used to simulate LW excitation and sensing. The other is solid mechanics controlled by fluctuation equations, which is used to simulate LW propagation. The temperature field is only a factor affecting LW propagation; that is, it is weakly coupled with the solid mechanics field and electrostatic field.

This paper adopts the COMSOL Multiphysics platform. Electrostatic and solid mechanics can be directly coupled. In addition, the thermal stress effect, piezoelectric effect, and temperature effect of PZTs can be integrated into the simulation model. The multiphysics simulation architecture of LW propagation with PZTs under temperature effect is shown in Figure 3.

In taking an aluminum plate as the research object, nine PZTs are arranged on an aluminum plate with an adhesive to simulate LW under temperature effect, as shown in Figure 4. The working condition of the simulation is shown in Table 2, described in detail as follows.

#### 3.1.1. D Geometry and Definitions

In this paper, the 3D geometry used includes the aluminum plate, the PZTs, and the adhesive layers. The size of the plate is 500 mm × 500 mm × 2 mm. The diameter and thickness of the PZTs are 8 mm and 0.48 mm, respectively. Nine PZTs are arranged symmetrically. The distance between two PZTs is 150 mm. The diameter and thickness of the adhesive layers are 8 mm and 0.08 mm, respectively.

There are two parts to define. One part is to define the voltage excitation signal that is modulated by a Hanning window, as shown in Equation (21). Parameters are set to *A* = 35 V, *f* = 200 kHz and *N* = 5. The other part is to define simulation temperature, reference temperature, and numerical model coefficients for material parameters under the influence of temperature.
(21)Ex=A⋅[1−cos(2πft/N)]⋅sin(2πft)⋅[t<(N/f)]
where *Ex* is the excitation signal, *A* is the amplitude, *f* is the central frequency, *t* is the wave propagating duration, and *N* is the number of cycles within the signal window.

#### 3.1.2. Material Parameters Numerical Model of Temperature Effect

According to the study in Section 2.2 and Section 2.3, the temperature influence on LW propagation can be equated to the effect of temperature on the material parameters of the structure, PZTs, and adhesive layers and the effect of thermal stress load on the material parameters of piezoelectric ceramics due to the inconsistent coefficient of thermal expansion. The numerical models of each part used in this paper are summarized, as shown in Table 3.

#### 3.1.3. Multiphysics Coupling under Temperature Effect

In solid mechanics, the default linear elastic materials, free and initial values are assigned to the aluminum structure, the adhesive layers, and the PZTs, while piezoelectric material is only assigned to the PZTs, and low reflection boundary is only assigned to the aluminum structure to suppress boundary reflection. In electrostatics, the upper surface of only one PZT is set as electric potential to apply a defined voltage signal, and the lower surface of all PZTs is set as ground. The voltage response signals of all PZTs can be received by defining the upper surfaces of all PZTs as boundary probes. The COMSOL will directly couple two physical fields in the calculation process by adding a piezoelectric effect to mulphysical fields and selecting solid mechanics and electrostatic.

For the calculation of thermal stress, it is necessary to add the thermal expansion to the nodes of linear elastic material and piezoelectric material, and then set the reference temperature to T0 and set the target temperature to Tem. It is considered that the temperature of structure, adhesive layers, and PZTs are the same, so it is not necessary to solve the solid heat transfer problem.

In solid mechanics physics, it is necessary to add rigid motion suppression to aluminum structure when calculating thermal stresses. The reason is that if there is no displacement constraint, the analysis will not converge without a unique solution, and the model will indicate “no solution found”. In the stationary study, it is necessary to find an equilibrium solution where the object is free to deform but not free to move or rotate, so the reaction forces must balance with each other. If no constraint is provided, the unbalanced forces will move or rotate the object, and the stationary solver cannot converge. COMSOL provides rigid motion suppression boundary conditions that can handle the missing displacement constraints and obtain the correct thermal stress results.

#### 3.1.4. Finite Element Meshes

Yang et al. [40] gave the relationship between the finite element size and the LW wavelength. The maximum mesh size recommended in the literature is 1/10 to 1/6 of the minimum wavelength. For LW at 200 kHz, the phase velocities of *S*_0_ mode and *A*_0_ mode on a 2 mm thick aluminum plate are 5382 m/s and 1731 m/s, respectively, and the corresponding wavelengths are 27 mm and 9 mm, respectively. Therefore, the maximum mesh size of the aluminum plate shall be less than 1.5 mm. Considering that the thickness of PZTs and adhesive layers are 0.48 mm and 0.08 mm, the mesh size of the PZTs and the adhesive layers are set to 1 mm and 0.5 mm, respectively. Due to the small mesh size, the model contains 1,890,000 domain elements, 553,000 boundary elements, and 3800 edge elements, with 5,200,000 degrees of freedom.

#### 3.1.5. Stationary and Time-Dependent Solver Settings

The solver includes the stationary study for thermal stress simulation and the time-dependent study for LW propagation. First, the stationary study is used to calculate the thermal stress of piezoelectric elements caused by inconsistent thermal expansion coefficient in step 1. Then, the thermal stress value is input into the parameter table. Finally, the time-dependent study is used to calculate LW propagation in step 2.

In the thermal stress simulation, the electrostatics and piezoelectric effect are disabled. In the LW propagation simulation, the thermal expansion and rigid motion suppression related to thermal stress simulation are disabled. If not disabled, the piezoelectric element will not only generate the LW signals but also be subjected to the thermal stress of the structure. At the beginning of the time-dependent study, the thermal stress at the coupling part between the piezoelectric element and the structure will act as a transient excitation to generate a wide-band voltage response signal.

When the simulation solver runs once, LW signals at one temperature level can be obtained. After several runs, LW signals at all temperature levels can be obtained. The simulation time step is set to 1 × 10^−7^ s, and the time range in the time-dependent study is from 0 s to 1.5 × 10^−4^ s.

### 3.2. Simulation Results

For LW at 200 kHz, the typical von Mises stress wave fields with the PZTs under temperature effect are shown in Table 4. It can be seen that both the *S*_0_ mode and *A*_0_ mode are excited normally. The *S*_0_ mode propagates faster than the *A*_0_ mode, and the amplitude of the *S*_0_ mode is weaker than that of the *A*_0_ mode. It can be seen that the amplitude increases with the increment of temperature by comparing the color bar of maximum stress at different temperatures. In order to better study the effect of temperature on the LW phase, the region in the red box is selected and enlarged. The wave fields at *t* = 3 × 10^−5^ s are given in Figure 5. It can be seen that the phase is delayed with the increment of temperature. The third wave packet just reaches the white line at 20 °C, exceeds the white line at −20 °C, and does not reach the white line at 60 °C.

The simulated LW signals under different temperatures are given in Figure 6, which are the LW signals of channel 5–6 when the center frequency of the excitation signal is 200 kHz. An enlarged view of *S*_0_ mode is given to observe the changes in phase and amplitude better. It can be seen that the amplitude increases, and the phase delays with the increment of temperature.

## 4. Experimental Verification of the Simulation Method

In order to verify whether the above simulation model can correctly and effectively simulate the LW propagation under the influence of temperature, experiments were conducted using an aluminum plate of the same size and material. The same adhesive and the PZTs were arranged at the same position. The LW signals at different temperatures were obtained. The differences between the experimental signals and the simulated signals were compared in terms of amplitude and propagation velocity to verify the correctness of the simulation method.

### 4.1. Experimental Setup

The geometric dimensions and material parameters of the aluminum plate, the adhesives, and the PZTs used in the experiment are the same as those of the simulation. PZT 5 is used to excite LW. The distance between two PZTs is 150 mm, as shown in Figure 7. The aluminum plate is fixed on an environmental test chamber THV1070W, which is used to provide the required temperature environment. The integrated SHM system [41] is used to excite and obtain LW signals. The experimental system of LW propagation in the aluminum plate under the influence of temperature is shown in Figure 8.

The whole range of temperature is from −20 °C to 60 °C, with the increment of 5 °C from −20 °C to 40 °C and the increment of 2 °C from 40 °C to 60 °C. The whole experimental process is about 6 h. There is no continuous signal acquisition in the process of temperature rise. The signal is acquired only when the temperature reaches the desired value. The excitation signal is a five-cycle sine burst modulated by a Hanning window with an amplitude of ±70 V. The central frequency of the LW excitation signal is 150 kHz and 200 kHz, respectively. The sampling rate is 10 MSamples/s. The signal acquired in the experiment is amplified by voltage, so the experimental signal cannot be directly compared with the simulation signal, and amplitude normalization is needed.

### 4.2. Experimental Results

The LW signals at 200 kHz under different temperatures are given in Figure 9. The first wave packet is the signal crosstalk, which can be ignored in the analysis. An enlarged view of the *S*_0_ mode is given to observe the changes in phase and amplitude better. The amplitude increases, and the phase delays with the increase of temperature. Figure 10 shows the relationship between signal amplitude and temperature. The amplitude increases linearly with the increase in temperature.

### 4.3. Comparison between Simulation and Experiment

The comparison between simulation signals and experimental signals of channel 5–6 under the temperature of −20 °C, 20 °C, and 60 °C are shown in Figure 11. The experimental signals are amplified by a charge amplifier, but the simulation signals are not. So, in order to better compare the simulation signals with the experimental signals, the complex continuous Shannon wavelet transform is used for filtering, and then the amplitude of *S*_0_ mode is normalized. It can be found that the waveform of *S*_0_ mode matches well, while the amplitude and phase of *A*_0_ mode have small errors. The reason for the error may be that the wavelength of *A*_0_ mode is less than *S*_0_; therefore, the mesh size of *A*_0_ mode needs to be smaller to ensure sufficient accuracy.

Equations (22) and (23) are used to measure the changes in signal amplitude and phase at different temperatures [42]. The data fit cross zero was performed.
(22)$$\Delta \mathrm{Amp}=\frac{{\mathrm{Amp}}_{\mathrm{Tem}}-{\mathrm{Amp}}_{T0}}{{\mathrm{Amp}}_{T0}}\times 100\%$$
(23)$$\Delta c_{p}=\frac{-c_{p}^{2}}{l_{p}}\Delta t$$
where Amp_Tem_ is the amplitude of the LW signals at different temperatures, Amp_T0_ is the amplitude of the LW signals at −20 °C, *c*_p_ is the phase velocity, *l*_p_ is the distance of LW propagation, and Δ*t* is the time shift of the constant phase of the LW signals.

Figure 12 shows the quantitative variations of amplitude and phase velocity of *S*_0_ mode between simulation and experiment. Experiment 1 and experiment 2 represent channel 5–6 and channel 5–8, respectively. It can be seen that the amplitude increases, and the phase velocity decreases with the increment of temperature. However, there may be differences in the influence of temperature on material parameters between simulation and experiment, resulting in the amplitude variations rate of the experiment being greater than that of the simulation. The phase velocity variations match well between the simulation and experiment.

## 5. Conclusions

The paper aims to provide a contribution to the modeling of the LW propagation with the PZTs under temperature effect. Temperature mainly affects the propagation of LW from two aspects. On the one hand, temperature affects the material parameters. On the other hand, the influence of thermal stress on the piezoelectric constant is due to the inconsistent thermal expansion coefficient. The stationary study for thermal stress and the time-dependent study for LW propagation are established. The simulation results at −20 °C to 60 °C are obtained and compared with the experimental results. The results show that the waveform of *S*_0_ mode matches well, while the amplitude and phase of *A*_0_ mode have small errors. In addition, the influence of temperature on the LW between simulation and experiment is also consistent; that is, the amplitude increases, and the phase velocity decreases with the increase of temperature. However, there may be differences in the influence of temperature on material parameters between simulation and experiment, resulting in the amplitude change rate of the experiment being greater than that of the simulation.

However, there still exists some issues that will be studied in future work.

(1) LW propagation with the PZTs under load conditions will be considered and combined with the method proposed in this paper to realize the multiphysics simulation of LW propagation under temperature and load conditions.

(2) Complex structures and composite structures are widely used in the aerospace field. Therefore, it is very important to study the multiphysics simulation method of LW propagation in these structures under time-varying conditions.

## Figures and Tables

**Figure 1 sensors-22-06647-f001:**
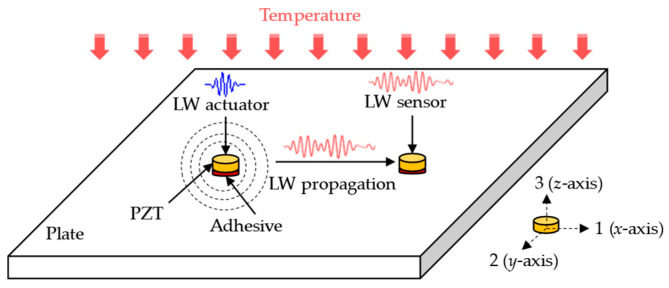
Schematic diagram of LW propagation with the PZTs under temperature effect.

**Figure 2 sensors-22-06647-f002:**
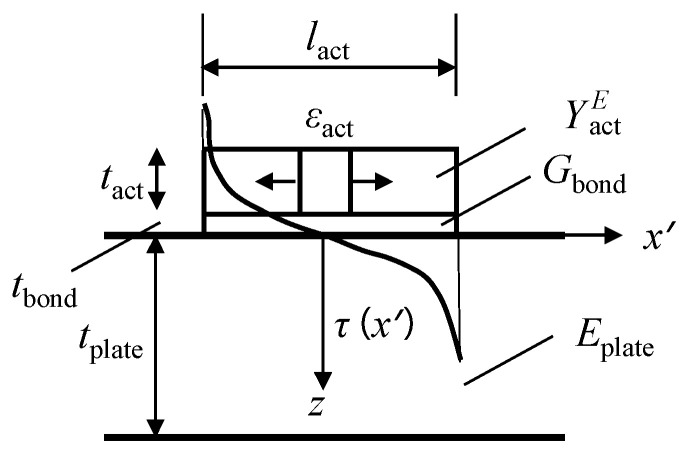
Schematic diagram of LW excitation [35].

**Figure 3 sensors-22-06647-f003:**
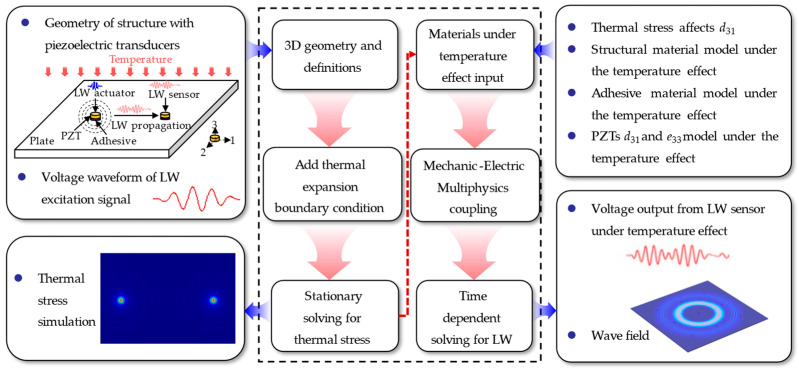
Architecture of multiphysics simulation method of LW under temperature effect.

**Figure 4 sensors-22-06647-f004:**
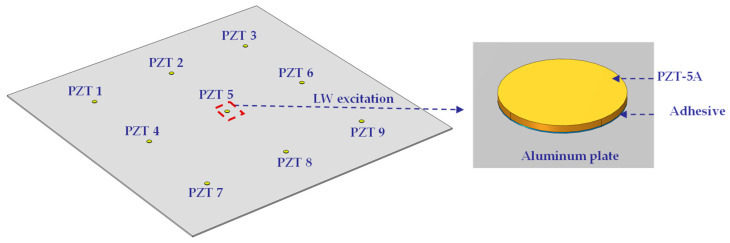
LW propagation under temperature effect simulation model.

**Figure 5 sensors-22-06647-f005:**
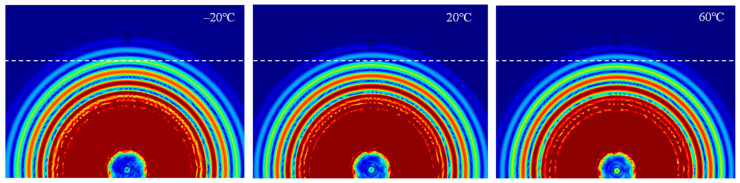
The effect of temperature on LW phase.

**Figure 6 sensors-22-06647-f006:**
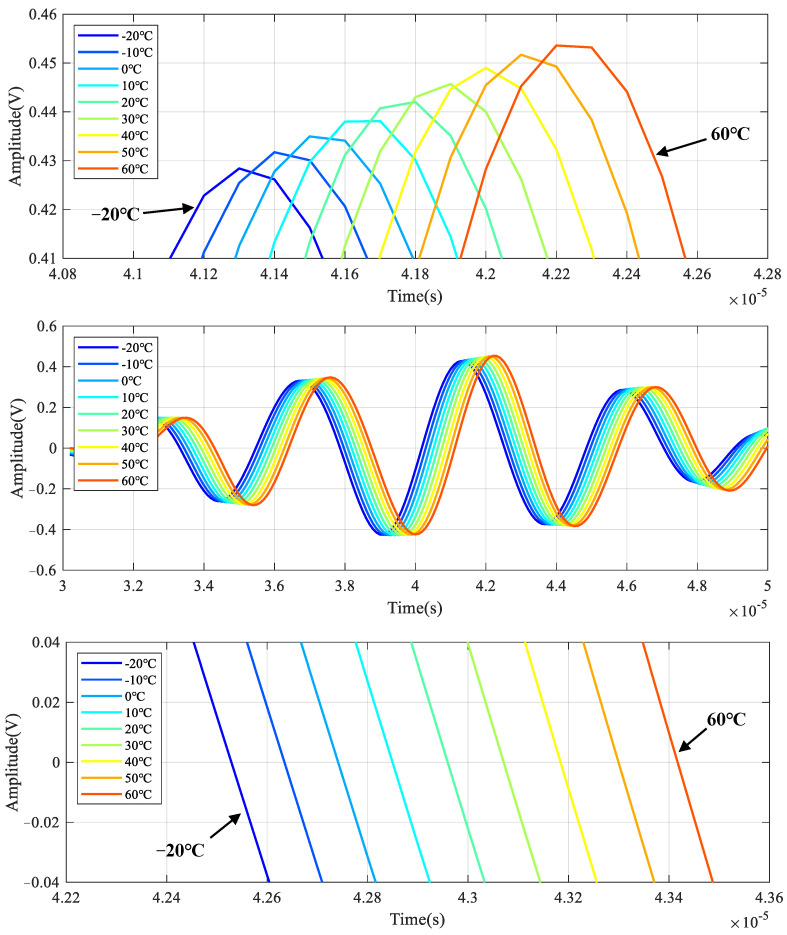
Typical LW response signal results of simulation under temperature effect.

**Figure 7 sensors-22-06647-f007:**
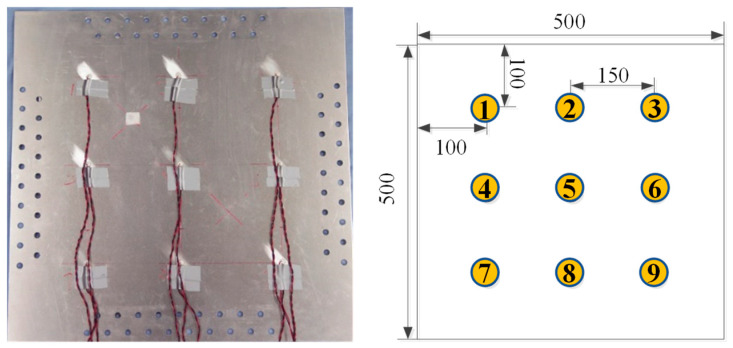
The aluminum plate with the PZTs and placement of the PZTs network.

**Figure 8 sensors-22-06647-f008:**
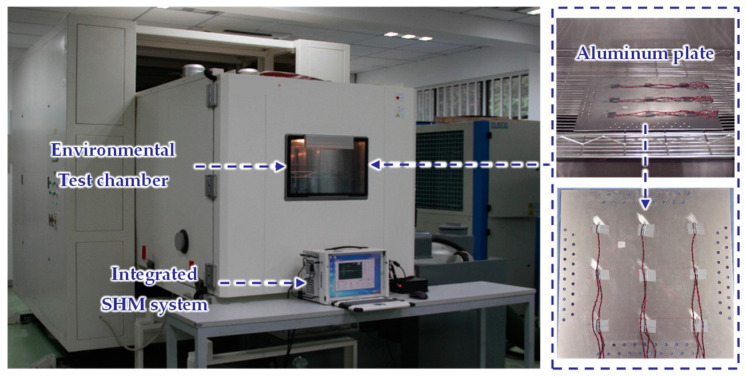
Experimental system of temperature influence on LW.

**Figure 9 sensors-22-06647-f009:**
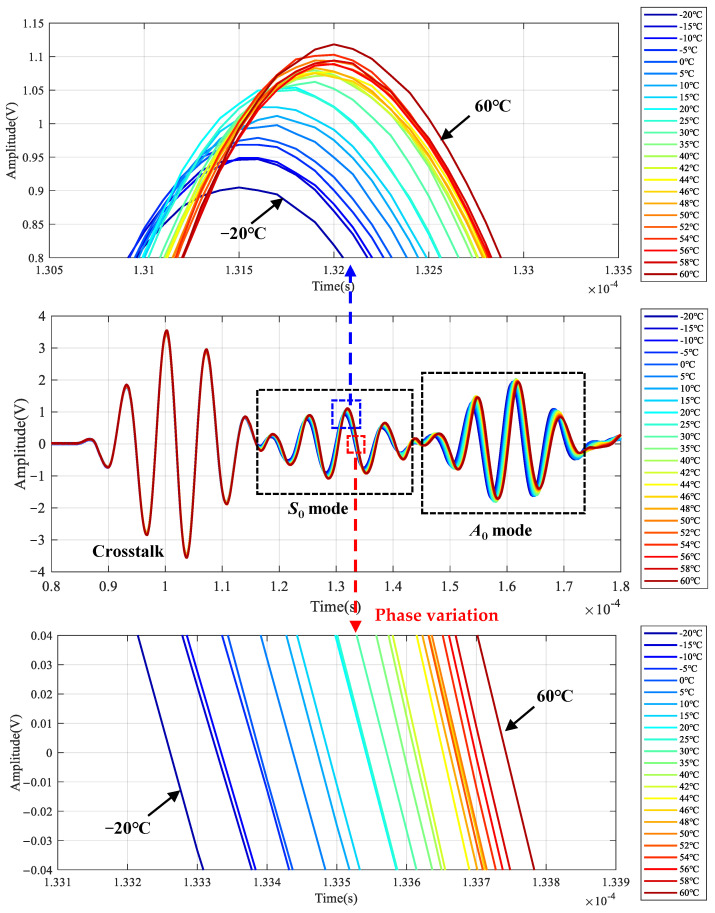
LW signals of central frequency 200 kHz at all temperatures.

**Figure 10 sensors-22-06647-f010:**
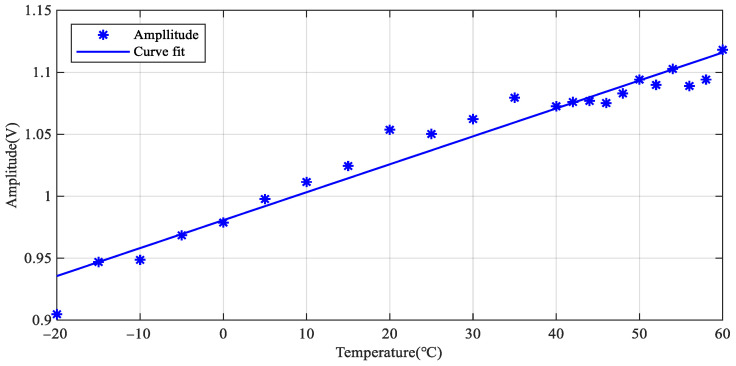
Amplitude change of LW signals at all temperatures.

**Figure 11 sensors-22-06647-f011:**
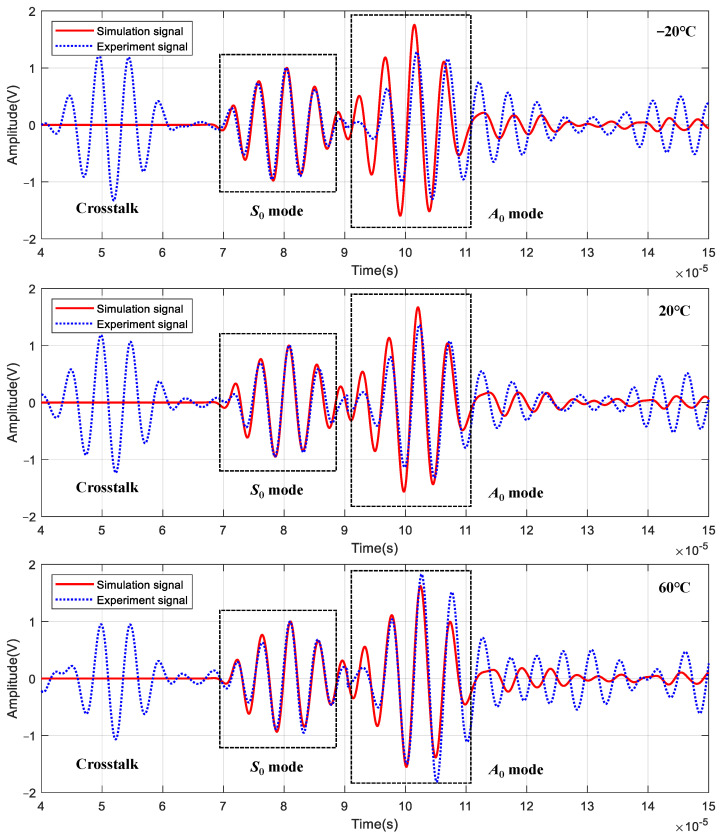
Comparison of LW signals between simulation and experiment.

**Figure 12 sensors-22-06647-f012:**
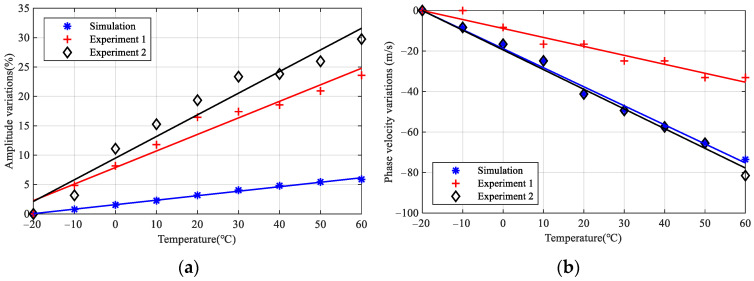
Comparison for variations of *S*_0_ mode between simulation and experiment. (**a**) Variations of amplitude; (**b**) variations of phase velocity.

**Table 1 sensors-22-06647-t001:** Summary of Lamb Wave theory model and key parameters influenced by temperature.

LW Propagation Characteristics	Expressions	Temperature Effects
Propagation velocity	$c_{L}=\sqrt{\frac{Y_{\mathrm{plate}}^{E}(1-\nu_{\mathrm{plate}})}{\rho_{\mathrm{plate}}(1+\nu_{\mathrm{plate}})(1-2\nu_{\mathrm{plate}})}}$ $c_{T}=\sqrt{\frac{Y_{\mathrm{plate}}^{E}}{2\rho_{\mathrm{plate}}(1+\nu_{\mathrm{plate}})}}$	Temperature affects the propagation velocity of LW by influencing the elastic modulus $Y_{\mathrm{plate}}^{E}$, density *ρ*_plate_, and Poisson’s ratio *ν*_plate_ of the structure.
Response amplitude	$V_{\mathrm{out}}\left(t\right)=d_{31}^{\mathrm{act}}C_{\mathrm{act}}\left(\Gamma \right)C_{\mathrm{sen}}\left(\Gamma \right){\left[\frac{d_{31}}{e_{33}s_{13}\left(1-\nu_{\mathrm{act}}\right)}\right]}_{\mathrm{sen}}V_{\mathrm{in}}\left(t\right)$ $\Gamma^{2}=\frac{G_{\mathrm{bond}}\;l_{\mathrm{act}}^{2}}{t_{\mathrm{bond}}}\left(\frac{1}{Y_{\mathrm{act}}^{E}t_{\mathrm{act}}}+\frac{\alpha }{E_{\mathrm{plate}}\;t_{\mathrm{plate}}}\right)$	Temperature affects the LW amplitude by influencing piezoelectric coefficient *d*_31_ including the effects of thermal stress, the dielectric constant *e*_33_ and elastic flexibility coefficient *s*_13_ of the PZT, shear modulus *G*_bond_ and shear-lag constant of the adhesive layer.

**Table 2 sensors-22-06647-t002:** Working condition of LW propagation simulation under temperature effect.

Structure	Geometry	Excitation Signal Frequency	Temperature
2024 Aluminum plate	500 mm × 500 mm × 2 mm (length × width × thickness)	150 kHz, 200 kHz	−20 °C to 60 °C

**Table 3 sensors-22-06647-t003:** Material properties of aluminum plate, adhesive, and PZT (PZT-5A).

Material	Parameter	Value
2024 Aluminum plate	Elastic modulus	$E_{\mathrm{plate}}\left(\Delta T\right)\left(\mathrm{GPa}\right)=73.5-0.06\Delta T$
	Poisson’s ratio	$v\left(\Delta T\right)=0.344+5.13\times 10^{-5}\Delta T$
	Density	2700 (kg/m^3^)
	Coefficient of thermal expansion	23.1 × 10^−6^(/K)
Adhesive	Shear modulus	$G_{\mathrm{bond}}\left(\Delta T\right)\left(\mathrm{GPa}\right)=3.61-0.01\Delta T$ 0.3
	Poisson’s ratio	
	Density	1110 (kg/m^3^)
	Coefficient of thermal expansion	54 × 10^−6^ (/K)
PZT-5A	Piezoelectric constant	$\begin{array}{l} d_{31}(\Delta T,\Delta \sigma_{T})=-167.7-0.194\times \Delta T\\ -167.7\left(-1.1\times 10^{-5}\times \Delta \sigma_{T}{}^{2}+4.2\times 10^{-3}\times \Delta \sigma_{T}\right) \end{array}$
	Relative permittivity	$e_{33}\left(\Delta T\right)=4.12\times \Delta T+2155$
	Coefficient of thermal expansion	3 × 10^−6^ (/K)

**Table 4 sensors-22-06647-t004:** Typical wave fields of LW propagation with the PZTs under temperature effect.

	3 × 10^−5^ s	4.5 × 10^−5^ s	6 × 10^−5^ s
−20 °C	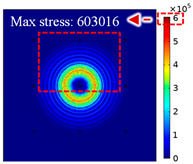	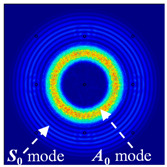	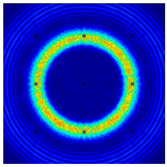
20 °C	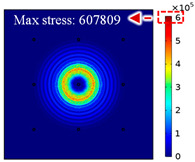	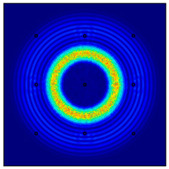	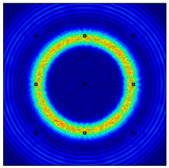
60 °C	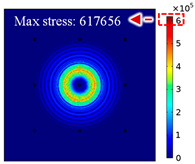	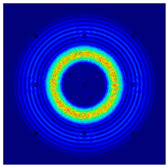	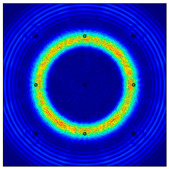

## Data Availability

Not applicable.

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
