# Peer review of "Mechanic-Electric-Thermal Directly Coupling Simulation Method of Lamb Wave under Temperature Effect"

_sensors, 2022, doi:10.3390/s22176647_

Round 1

Reviewer 1 Report

Dear Authors,

This paper presents a finite element method of modeling the temperature effect on the Lamb Wave excitation and propagation. As mentioned by the authors, this will help expand the database for SHM without expensive experiments. The created model is well supported by the literature data. The reviewer believes it is in a good shape. However, the reviewer had the following questions. Hopefully, by answering them and adding more details, the paper will be more clear and easy for future readers to follow.

1.       In Table 3, only the shear modulus of the adhesive material is used. Is Young’s modulus not needed? Is the adhesive material modeled as a linear elastic material in COMSOL? If it is modeled as a linear elastic material, are values other than shear modulus set to be 0?

2.       In Line 317, the modeling method of the transmitter PZT is described. However, the reviewer did not find the information of receiver PZTs. Are the receiver PZTs included in the thermal study also? Are the receivers set to detect voltage in the COMSOL models? It would be helpful if the authors provide more details on the receiver PZT modeling.

3.       In Table 4, what is the wavefield? Is that displacement, velocity, or something else? Is that the absolute displacement or a particular component of displacement, like ux, uy, or uz?

4.       In Line 428, when it comes to the complex continuous Shannon wavelet transform, the reviewer did not follow it. What is the function of this transformation? Does that transformation adjust the amplitude of S0 and A0 proportionally? It would be beneficial if the authors could provide more details since matching the simulation results to the experimental results are critical. 

Reviewer 2 Report

This paper proposes a multiphysics simulation method for LW propagation with the  PZTs under temperature effects. Valuable results are obtained however, minor changes are recommended before publishing as follows:

1. In the abstract, please indicate the comparison results between the simulation and experimental at the tested temperatures. 

2. Some references can be added in the introduction to illustrate the effects of temperature on the lamb wave :

“Evaluation of effect changing temperature on lamb-wave based structural health monitoring”

“Environmental and operational conditions effects on Lamb wave based structural health monitoring systems: A review”

3. In the introduction, lines 100-105 are not necessary and can be omitted. 

4. In section 2, on which base the example in line 107 is chosen. Please refer to the reference.

5. The resolution of Figure 2 should improve.

6. The equations format should improve throughout the manuscript.

7. The conclusion does not describe the results of the manuscript and it is too long.

8. The English language needs to be improved and some repeated information must be merged.

Reviewer 3 Report

The article is devoted to modeling the propagation of LW with the help of PZTs at different temperatures. It combines thermomechanical and electromechanical communication.

The article contains paragraphs describing the further presentation of the material. In my opinion, this is unnecessary. For example:

"The reminder of this paper is given as follows. In Section 2, the two key mechanisms of temperature influence on LW propagation are discussed, and the influence of temperature on the PZTs is also established. The modeling method and simulation results of LW propagation with the PZTs under temperature effect are given in Section 3. Then, LW propagation experiments at different temperatures are carried out on aluminum plate to verify the simulation method. The last Section is devoted to conclusions."

or

"3.1.2. Materials 

The material parameters involved in the model include the aluminum plate, the adhesive layers and the PZTs. The effect of temperature on these material parameters will be given in Section 3.2."

It follows from the work that only two experiments were given. And why so little, because the experiment takes only 3 hours. A large number of experiments will give a more plausible picture

During the experiment, was the signal LW supplied continuously when the temperature changed, or only when the desired value was reached?

As the temperature increases, the epoxy adhesive becomes more plastic. How was this taken into account in the model calculations?

In Figure 13, when comparing the experiment and the model at the beginning, the maximum waves for temperatures 20 and 40 practically coincide. But then the extremes shift, which is why.

The same picture shows why there is a "failure" on experimental data when comparing the 5th and 6th waves.

Reviewer 4 Report

Reviewed paper is related influence of temperature on Lamb wave propagation. This topic is important but relatively well investigated in the literature in the case of isotropic materials studied in this manuscript. Authors performed numerical and experimental research. In my opinion this paper bring nothing interesting to the field of Lamb waves propagation and SHM. Even for conducted research and obtained data there is lack of some sophisticated analysis. Such a research could be interested for composite materials. This manuscript in its present form could be published as report from measurements or after some improvements as conference paper but not as regular journal paper.

Please find my comments that will allow to improve this paper for possible publication in the future:

  1. Signals presented in fig. 13 – The time length is limited only to directly propagated A0, S0 wave modes. Good agreement of numerical and experimental results is always achieved for direct waves. However, longer time duration of signal, containing waves reflection (utilised in damage detection) show very often some discrepancies. Please compare signal with more samples (longer in time).
  2. Small range of temperatures (20 - 60) without negative temperatures
  3. Fig. 5, 6, 7 brings nothing interesting to the manuscript. It is scientific Journal where the scientific results should be published. Such a figures could be placed in report or some tutorial.
  4. Similarly fig. 4. – please remove the window of COMSOL related to options and leave only the part with view of the plate - I think most important in this figure.
  5. Results in table 4 and fig. 8. Please perform some analysis of these figures. Really I do not see any large differences.
  6. Fig. 12 Please plot the amplitudes in function of temperature.

Round 2

Reviewer 3 Report

Ok

Reviewer 4 Report

This manuscript was improved according to my comments and could be publish in present form. However, my general assessment of manuscript level is low considering the novelty and average taking into account its significance to the reader.